# Viral Evolution and Immunology of SARS-CoV-2 in a Persistent Infection after Treatment with Rituximab

**DOI:** 10.3390/v14040752

**Published:** 2022-04-03

**Authors:** Nathalie Van der Moeren, Philippe Selhorst, My Ha, Laura Heireman, Pieter-Jan Van Gaal, Dimitri Breems, Pieter Meysman, Kris Laukens, Walter Verstrepen, Natasja Van Gasse, Benson Ogunjimi, Kevin K. Arien, Reinout Naesens

**Affiliations:** 1Laboratory of Microbiology, Ziekenhuis Netwerk Antwerpen, 2050 Antwerp, Belgium; laura.heireman@zna.be (L.H.); walter.verstrepen@zna.be (W.V.); natasja.vangasse@zna.be (N.V.G.); reinout.naesens@zna.be (R.N.); 2Department of Biomedical Sciences, Institute of Tropical Medicine Antwerp, 2000 Antwerp, Belgium; pselhorst@itg.be (P.S.); karien@itg.be (K.K.A.); 3Centre for Health Economics Research & Modelling Infectious Diseases (CHERMID), Antwerp Centre for Translational Immunology and Virology (ACTIV), Vaccine & Infectious Disease Institute (VAXINFECTIO), Department of Paediatrics, Antwerp University, 2610 Antwerp, Belgium; my.ha@uantwerpen.be (M.H.); pieter.meysman@uantwerpen.be (P.M.); kris.laukens@uantwerpen.be (K.L.); benson.ogunjimi@uantwerpen.be (B.O.); 4Departement of Nephrology, Ziekenhuis Netwerk Antwerpen, 2050 Antwerp, Belgium; pieter-jan.vangaal@zna.be; 5Department of Haematology, Ziekenhuis Netwerk Antwerpen, 2050 Antwerp, Belgium; dimitri.breems@zna.be

**Keywords:** SARS-CoV-2, persistent infection, immunocompromised host, variants of concern (voc), viral evolution

## Abstract

Background. Prolonged shedding of SARS-CoV-2 in immunocompromised patients has been described. Furthermore, an accumulation of mutations of the SARS-CoV-2 genome in these patients has been observed. Methods. We describe the viral evolution, immunologic response and clinical course of a patient with a lymphoma in complete remission who had received therapy with rituximab and remained SARS-CoV-2 RT-qPCR positive for 161 days. Results. The patient remained hospitalised for 10 days, after which he fully recovered and remained asymptomatic. A progressive increase in Ct-value, coinciding with a progressive rise in lymphocyte count, was seen from day 137 onward. Culture of a nasopharyngeal swab on day 67 showed growth of SARS-CoV-2. Whole genome sequencing (WGS) demonstrated that the virus belonged to the wildtype SARS-CoV-2 clade 20B/GR, but rapidly accumulated a high number of mutations as well as deletions in the N-terminal domain of its spike protein. Conclusion. SARS-CoV-2 persistence in immunocompromised individuals has important clinical implications, but halting immunosuppressive therapy might result in a favourable clinical course. The long-term shedding of viable virus necessitates customized infection prevention measures in these individuals. The observed accelerated accumulation of mutations of the SARS-CoV-2 genome in these patients might facilitate the origin of new VOCs that might subsequently spread in the general community.

## 1. Introduction

Prolonged shedding of SARS-CoV-2 has been repeatedly described in patients receiving immunosuppressive therapy [1,2,3,4,5]. The median duration of SARS-CoV-2 viral RNA shedding in upper respiratory samples of immunocompetent patients varies—among others with disease severity, host age and the infecting SARS-CoV-2 variant—but is mainly limited in time [6,7]. An overall median period of 12 days (95% CI 8–15) was concluded from a systematic review conducted in April 2020 when mainly the D614G non-variant of concern (VOC) was circulating in Europe [8].

However, in patients receiving immunosuppressive therapy, prolonged shedding of SARS-CoV-2 has been repeatedly described with nasopharyngeal (NP) specimens being reverse transcriptase quantitative polymerase chain reaction (RT-qPCR) positive for SARS-CoV-2 for over 150 days [6,9].

Important clinical implications rise as these persistent infections might be associated with adverse outcome [9]. Furthermore, the long-term shedding of viable virus implies a need for customized infection prevention measures in and outside the hospital [1]. Foremost, an accelerated accumulation of mutations of the SARS-CoV-2 genome is observed in patients with a persistent SARS-CoV-2 infection [4]. The partial immune response in these patients might provide the perfect conditions for new VOCs that subsequently can spread in the general community.

Here we describe the clinical course, immunologic response and viral evolution of a haematological patient who had received treatment with rituximab, a humanized chimeric monoclonal antibody against the pan-B-cell marker CD20 used in the treatment of B-cell malignancies and various autoimmune diseases [10,11]. The patient remained SARS-CoV-2 RT-qPCR positive for 161 days.

## 2. Results

### 2.1. Case Report

A 60-year-old man presented at the emergency room (ER) of a Belgian hospital with high fever (39.2 °C) and a productive cough. His medical history consisted of type 2 diabetes mellitus, obesity and peripheral vascular insufficiency. Furthermore, he had been diagnosed with a mantle cell lymphoma with paraneoplastic glomerulonephritis for which he needed renal replacement therapy (intermittent haemodialysis). Therapy for the mantle cell lymphoma consisted of six courses of rituximab and bendamustin during a seven-month period. After a PET-CT showed complete remission and his kidney function recovered to a stable eGFR of 25–30 mL/min/1.73 m^2^ (CKD-EPI formula), maintenance therapy with rituximab had been initiated. The second dose was administered seven days before presentation at the ER (Figure 1). The laboratory showed leukopenia (1.35 × 10^9^/L) with neutropenia (0.96 × 10^9^/L) and remarkable lymphopenia (0.01 × 10^9^/L) (Table 1). SARS-CoV-2 RT-qPCR performed on a NP swab was positive with a cycle-threshold (Ct) of 16.3 corresponding to a semi-quantitative estimation of the viral load of 5.14 × 10^8^ virus copies/mL. A CT-scan on day one showed bilateral consolidations and ground-glass opacifications consistent with the image of COVID-19-associated pneumonia. A six-day course of meropenem (1000 mg twice daily) was initiated to treat possible bacterial pulmonal superinfection. On day one of hospitalisation, the patient developed hypoxemia, requiring nasal oxygen therapy until day five. Cultures of sputum and blood were negative for the growth of bacteria or fungi. Between day one and three, three dosages of convalescent plasma were administered. Dexamethasone treatment (6 mg once daily) was started on day five and administered until day ten. The patient progressively recovered and was discharged on day ten (Figure 1).

On day 21, the patient was referred to the emergency department because of fever (temperature of 39 °C). Chest X-ray showed a novel consolidation in the right inferior lobe. SARS-CoV-2 RT-qPCR on the obtained NP swab was positive with a Ct-value of 20.8. Immunoassay for SARS-CoV-2 anti-nucleocapsid immunoglobulins (N-antibodies) was negative. The patient was discharged with a five-day antibiotic course (moxifloxacin 400 mg once daily) and fully recovered. Because of the strongly positive RT-qPCR, the patient was advised to resume self-isolation at home.

A follow-up NP sample obtained on day 37 remained positive for SARS-CoV-2 RNA. At this point it was decided to perform a broad evaluation of the patient including NP, sputum, faecal and plasma specimen for RT-qPCR, genome sequencing and viral culture, serum samples for SARS-CoV-2 serology and peripheral blood for immunologic analysis. The samples were, due to practical concerns, obtained on day 67. SARS-CoV-2 RT-qPCR on the NP and sputum sample were positive with a Ct of 19.0 and 24.8, respectively. No SARS-CoV-2 RNA was detected in urine or plasma although a faecal sample was found borderline positive (Ct 38.1). Analysis of peripheral blood showed no detectable B-cells and severe T-cell depletion. Immunoassay for N-antibodies was negative. SARS-CoV-2 spike protein immunoglobulins (S-antibodies), however, were not routinely determined at our laboratory at this time. (Table 1)

Between day 26 and 161, the patient remained asymptomatic and self-isolated at home. NP swabs obtained on day 111 and 124 showed stable Ct-values (21.8 and 19.7, respectively) (Table 1).

On day 137, the Ct-value on the NP swab rose to 28.6. Serum at this time showed a borderline positive N-antibody titre (Index 1.2) and an S-antibody titre above the upper limit of detection (of 250 IU/mL).

A further increase in Ct-value to 35.2 was seen on day 152. Although RT-qPCR was still positive at this time, the high Ct-value was considered to correspond with a low viral load. As the patient also had a high S-antibody titre, he was considered to be no longer infectious. After a confirmatory RT-qPCR on day 161 rendered a comparable Ct-value, the home quarantine of the patient was lifted, and he was able to resume his job.

### 2.2. Virologic Analysis

Whole genome sequencing (WGS) of viral sequences obtained from NP and sputum samples collected at different time points showed that the patient was persistently infected by the same virus belonging to the wild type (non-VOC) SARS-CoV-2 clade 20B/GR. At day 37, a total of 14 nucleotide differences were observed relative to the Wuhan-Hu-1 reference gradually evolving to 24 differences at day 137 (Table 2). This corresponds to three mutations per month, which is higher than the estimated evolutionary rate in the infected population of roughly 1.5 substitutions per month for non-VOCs (0.58 × 10^−3^ subs/site/year (95% CI: 0.51−0.65 × 10^−3^)) [12].

Specifically, between day 37 to 67, the virus evolved deletions in the N-terminal domain (NTD) of its spike protein (S) at amino acid position 241–243, and later, between day 111 to 124, also at position 143–144 (Figure 2. These mutations preserve the reading frame and were previously reported in chronic, immunocompromised patients, and occur at a global scale in the GISAID database [1,5,13]. This independent evolution of recurrent NTD deletions in SARS-CoV-2 has therefore been suggested as a mechanism to introduce variation at specific antigenic sites and drive antibody escape [14]. Interestingly Δ143 and Δ144 are signature mutations in VOC alpha, and variants of interest (VOI) eta, theta, and iota, while Δ242 and Δ243 developed in VOC beta and Δ243 in VOI theta [15,16]. In addition to these NTD deletions, our patient’s virus developed the N501Y mutation between day 67 to 111 in the receptor binding domain (RBD) of S. This mutation is believed to increase the binding affinity of S to the ACE2 receptor, and hence to play an essential role in the increased transmissibility of the different SARS-CoV-2 lineages in which it convergently evolved thus far (i.e., alpha, beta, gamma, theta). Notably, this mutation was first detected in the alpha lineage in September 2020, which is hypothesized to have evolved in a chronically-infected immunocompromised individual [17]. In this patient, convalescent plasma treatment was shown to be responsible for the selection of a variant with decreased susceptibility to neutralizing antibodies. In our patient, however, the role of convalescent plasma treatment in the development of the NTD deletions and N501Y remains unclear as it was administered in the first days of hospitalisation, whereas the earliest sequence data could be obtained on day 37. Finally, three other mutations, namely T95I, A475S, and F490S emerged in the spike protein over time. T95I has been described in VOC delta, iota, and kappa. The RBD mutation F490S, on the other hand, is a signature mutation in VOC lambda and has been shown to confer resistance to vaccine-induced antisera and monoclonal antibodies in respective pseudoviral assays and chimeric virus experiments [17,18]. Potentially, the emergence of this mutation between day 111 and 124 coincided with the appearance of anti-SARS-CoV-2 antibodies first measured on day 137, but which probably arose earlier, as suggested by the high concentration of anti-S antibodies at that time point (above the upper limit of detection).

Lastly, culture of the nasopharyngeal swab specimen taken at day 67 showed viral growth on Vero-E6 TMPRSS2 cells already after one day of incubation, indicating the virus was still highly infectious at this time point.

### 2.3. Immunophenotyping

Peripheral blood mononuclear cells (PBMC) analysis on day 137, showed a decreased frequency of B-cells and T-cells when compared to samples of healthcare workers obtained a comparative amount of time after SARS-CoV-2 infection (*p* < 0.01) (Figure 3) Myeloid (= conventional) dendritic cells (mDCs), in contrast, were found to be increased in frequency (*p* < 0.01). Non-specific (thus not specifically against SARS-CoV-2) CD4^+^ and CD8^+^ T-cells showed signs of activation with high expression of OX40, a good indicator for antigen specific T-cell activation. TIGIT and Fas were significantly upregulated in respective CD4^+^OX40^+^ and CD8^+^OX40^+^ T-cells of patients compared to the controls (Figure 3 and Figure 4).

## 3. Materials and Methods

### 3.1. SARS-CoV-2 RT-qPCR

SARS-CoV-2 reverse transcriptase quantitative polymerase chain reaction (RT-qPCR) was performed with primers and probe directed to the N1-target of the SARS-CoV-2 gene (CDC 2019-Novel Coronavirus (2019-nCoV) Real-Time RT-qPCR Diagnostic Panel, CDC, Atlanta). Extraction was performed with MagNaPure 96 (Roche, Basel, Switzerland), amplification with the Lightcycler 480 (Roche, Basel, Switzerland). A semi-quantitative estimation of viral loads from Ct-values was made using a standard curve based on the analysis of standardised samples from the Belgian national reference laboratory (National Reference Laboratory UZ Leuven and KU Leuven, Leuven, Belgium).

### 3.2. SARS-CoV-2 Whole Genome Sequencing (WGS)

WGS was performed on an Oxford Nanopore MinION device using R9.4 flow cells (Oxford Nanopore Technologies, Oxford, UK) after a multiplex qPCR with an 800 bp SARS-CoV-2 primer scheme as previously described [19]. Sequence reads were basecalled in high accuracy mode and demultiplexed using the Guppy algorithm v3.6. Reads were aligned to the reference genome Wuhan-Hu-1 (MN908947.3) with Burrows-Wheeler Aligner (BWA-MEM), and a majority rule consensus was produced for positions with ≥100 x genome coverage, while regions with lower coverage, were masked with N characters. Sequence alignment was performed using MAFFT v7. Clade assignment and amino acid and nucleotide comparison to the reference genome were performed using NextClade v0.7.2, (Basel, Switzerland) [20].

### 3.3. Virus Culture

Virus culture was performed by incubating a serial dilution of nasopharyngeal samples on 18,000 VeroE6-TMPRSS2 cells per well after 2 h of spinoculation at 2500× *g* and 25 °C and following up cytopathic effect. Assay medium consisted of EMEM (Lonza, Verviers, Belgium) supplemented with 2 mM L-glutamine, 2% fetal bovine serum, and penicillin—streptomycin (Lonza, Verviers, Belgium).

### 3.4. Immunologic Analysis

#### 3.4.1. SARS-CoV-2 Serology

SARS-CoV-2 anti-nucleocapsid and spike-IgG in plasma were determined with the Elecsys Anti-SARS-CoV-2 immunoassay (Roche, Basel, Switzerland) in accordance with the manufacturer instructions.

#### 3.4.2. Immunophenotyping

Peripheral blood mononuclear cells (PBMC) of the patient were obtained at day 67 and day 137. The sample of day 67 did not contain sufficient PBMC for analysis. High-dimensional mass cytometry was used to analyze PBMCs of the patient and—as a comparison—of three healthcare workers who had been diagnosed around the same time.

PBMCs were stimulated with PepTivator Prot-S1 (Miltenyi Biotec, Bergisch Gladbach, Germany) and customised MHC-specific (JPT Peptide Technologies, Berlin, Germany) SARS-CoV-2 peptide pools for 16 h at 37 °C and 5% CO_2_. PepTivator^®^ Prot-S1 is a pool of lyophilized peptides, covering the N-terminal S1 domain of the surface glycoprotein (“S”), while MHC-specific peptides are pools of peptides specific to the major histocompatibility complexes I and II in immune cells. Negative controls were prepared in the same condition but without peptide stimulation. After incubation, cells were labelled with surface and intracellular markers according to the Maxpar Cell Surface Staining with Fresh Fix protocol (Fluidigm Corp., South San Francisco, CA, USA). Briefly, cells were washed with Cell Staining Buffer (Fluidigm Corp., South San Francisco, CA, USA) and then stained with Cell-ID Intercalator-103 Rh (Fluidigm Corp., South San Francisco, CA, USA) for viability. Cells were then stained with metal-tagged antibodies. All metal-tagged antibodies were purchased pre-conjugated from Fluidigm Corp., South San Francisco, CA, USA. Subsequently, cells were fixed in formaldehyde (Thermo Scientific, Waltham, MA, USA) for 10 min at room temperature and incubated with Cell-ID Intercalator-Ir (Fluidigm Corp., South San Francisco, CA, USA) for 1 h at room temperature. Prior to data acquisition, cells were pelleted and re-suspended at 1 × 10^6^ cells/mL in Cell Acquisition Solution (Fluidigm Corp., South San Francisco, CA, USA). Helios mass cytometer (Fluidigm Corp., South San Francisco, CA, USA) was used for data acquisition. The instrument was tuned by optimizing the nebulizer, makeup gas, current, and detector voltage according to the manufacturer’s guideline. The injection speed (or flow rate) was set to 5 × 10^−7^ L/s, and push length was 13 µs by default.

Cytometric data were analysed on FlowJo (FlowJo, LLC, Williamson Way Ashland, OR, USA) and the open source statistical software R. Visualisation was performed with R and GraphPad Prism (GraphPad Software, San Diego, CA, USA).

## 4. Discussion

We describe the clinical course, immunologic response and viral evolution of a patient with a mantle cell lymphoma in complete clinical remission who received maintenance therapy with rituximab and remained SARS-CoV-2 RT-qPCR positive during a 161-day period.

Multiple case reports of prolonged viral shedding in patients receiving immunosuppressive therapy have been published [1,2,3,4,5]. Epasse et al., among others, described two SARS-CoV-2 infections in patients receiving rituximab, both with a fatal outcome [8]. Despite two episodes of clinical illness, this patient fully recovered and remained asymptomatic for 135 days of the 161 days SARS-CoV-2 RNA was detectable. In contrast to the previous reported cases, immunosuppressive therapy was discontinued from the moment of SARS-CoV-2 diagnosis, and there was no active underlying condition. A progressive increase in Ct-value, coinciding with a progressive rise in lymphocyte count, was seen from day 137 onward.

The accumulation of mutations and deletions observed within the spike NTD and RBD are in concordance with previously published case reports [1,2,3,13]. Meta-analysis of whole genome sequencing data of nine immunosuppressed patients demonstrated accelerated viral evolution in these individuals compared to the background evolutionary rate in the general population and continuation of this evolution during treatment with monoclonal antibodies or convalescent plasma [9]. Unfortunately, due to sample unavailability, we could not determine whether the administration of convalescent plasma drove the evolution in our patient. Interestingly, most of the mutations that occurred are signature mutations in new VOCs that later arose in the infected human population. This is in line with the hypothesis that viral evolution in immunocompromised patients may be an important factor in the emergence of such variants.

Virus culture at day 67 still showed highly infectious virus, which underlines the importance of implementing adequate infection control and quarantine measures to prevent community spread of this rapidly evolving virus. Most likely, this is due to the suboptimal immune response observed at this time point. Positive viral cultures up to 119 days after initial diagnosis have been described previously in immunocompromised patients [2,3].

Immunophenotyping showed a reduced frequency of peripheral B-cells and T-cells, and an increased frequency of dendritic cells when compared to healthcare workers at a comparable time after diagnosis. B-cell depletion and T-cell depletion were considered to be consequences of the treatment with rituximab [21].

Depletion of the antigen-presenting B-cells may diminish antigen presentation and consequently interfere with T-cell activation. However, this patient’s dendritic cells, another important antigen-presenting cell type, were increased in frequency. Although dendritic cells experience depletion and functional impairment in acute SARS-CoV-2 infection, our data show that they can recover during convalescence, particularly the myeloid dendritic cell population, as the blood was collected on day 167 when the patient was in convalescence [22]. Our findings agree with one study that also observed a higher percentage of mDCs in SARS-CoV-2 convalescent patients seven months post-infection compared to healthy donors [23]. Several studies have indicated that even in asymptomatic patients, viral shedding can be persistent for more than two months after diagnosis [24]. In the current case report, the presence of dendritic cells may have helped T-cell activation by presenting the persistent SARS-CoV-2 antigens in the absence of B-cells. Our data showed that CD4^+^ T-cells and CD8^+^ T-cells in the patient displayed signs of activation. Within the SARS-CoV-2-specific (presence of OX40 upregulation of stimulation with SARS-CoV-2 specific epitopes), OX40^+^CD4^+^ T-cells and OX40^+^CD8^+^ T-cells exhaustion markers, such as TIGIT and CD57, were upregulated [25]. This exhaustion is likely a consequence of the persistent SARS-CoV-2 presence and T-cell stimulation. In addition, the upregulation of TIGIT on OX40^+^CD4^+^ T-cells also signifies potential immunosuppressive activity [26].

## 5. Conclusions

Long-term SARS-CoV-2 persistence in immunocompromised individuals has important clinical implications, but halting immunosuppressive therapy might result in a favourable clinical course. The long-term shedding of viable virus necessitates customized infection prevention measures in these individuals. The observed accelerated accumulation of mutations of the SARS-CoV-2 genome in these patients might facilitate the origin of new VOCs that might subsequently spread in the general community.

## Figures and Tables

**Figure 1 viruses-14-00752-f001:**
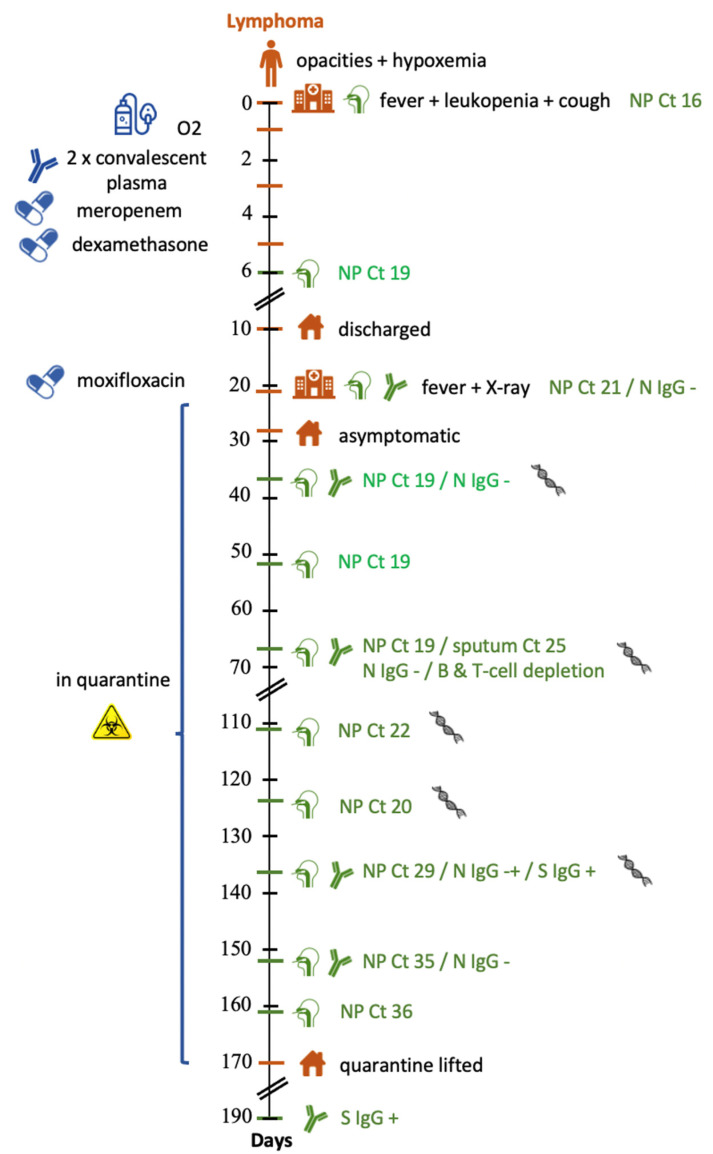
Schematic chronologic overview of the clinical course, sampling points and evolution of SARS-Cov-2 RT-qPCR Ct-value and antibodies. NP: nasopharyngeal specimen; Ct: SARS-CoV-2 RT-qPCR cycle threshold; N IgG: SARS-CoV-2 anti-nucleocapsid immunoglobulin; S IgG: SARS-CoV-2 spike protein immunoglobulin.

**Figure 2 viruses-14-00752-f002:**
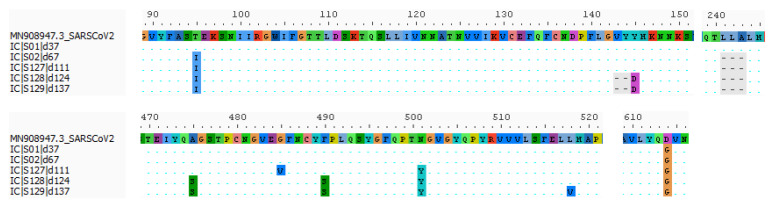
Amino acid changes in the S-gene of the patient’s SARS-CoV-2 genomes relative to the reference genome Wuhan-Hu-1. Deletions are marked in grey.

**Figure 3 viruses-14-00752-f003:**
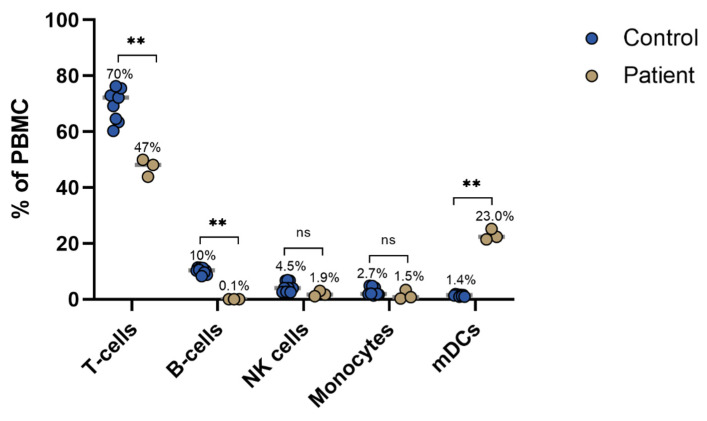
Frequencies of major immune subsets. Significance levels analysed by the Mann–Whitney test: ns = *p* > 0.1, ** = *p* ≤ 0.01. Gray lines indicate mean values. PBMC: peripheral blood mononuclear cells.

**Figure 4 viruses-14-00752-f004:**
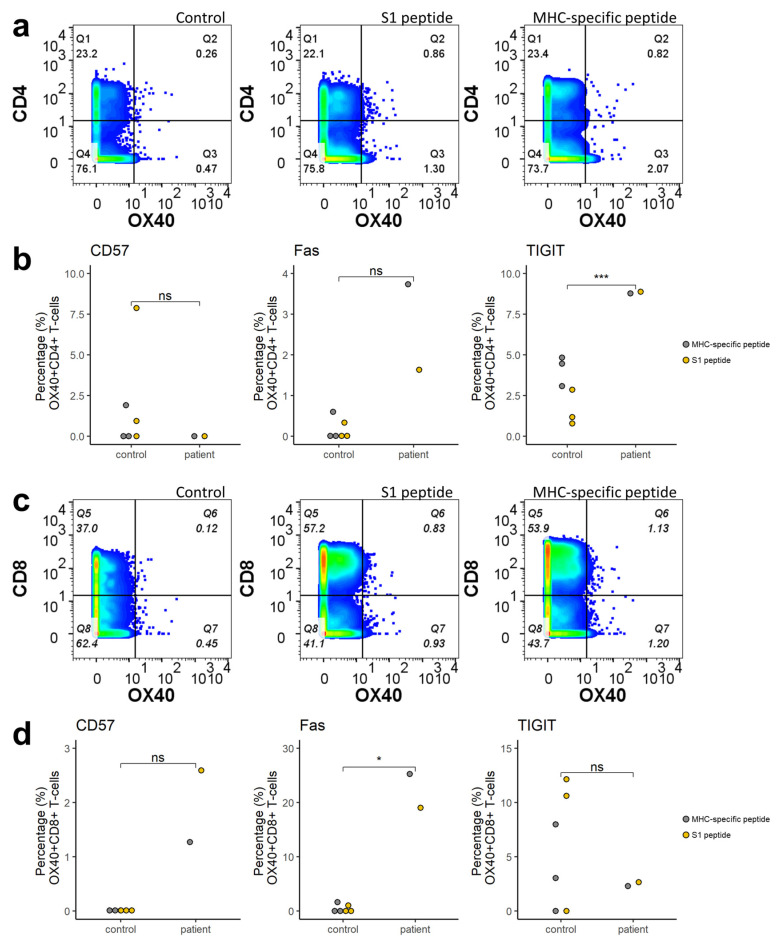
(**a**,**c**) CD4^+^ and CD8^+^ T-cells from the patient displayed upregulation in OX40 expression when stimulated with S1 and MHC-specific peptides compared to unstimulated cells. (**b**,**d**) Expressions of functional markers in antigen-specific OX40^+^CD4^+^ and OX40^+^CD8^+^ T-cells. Significance levels analysed by the Mann–Whitney test: ns (not significant) = *p* > 0.1, * = *p* ≤ 0.1, *** = *p* ≤ 0.001. MHC: major histocompatibility complex.

**Table 1 viruses-14-00752-t001:** Overview of the evolution of SARS-CoV-2 RT-qPCR Ct-values, haematology and SARS-CoV-2 antibodies.

Day	RT-qPCR Ct-Value *	Viral Load (Copies/mL) **	WGS ***	Lymphocytes (10^9^/L)	B-Lymphocytes (10^9^/L)	T-Lymphocytes (10^9^/L)	N-Prot Ab (Index) ****	S-Prot Ab (U/mL) *****
0	16.27	5.14 × 10^8^		0.01				
7	19.90	4.15 × 10^7^		0.04				
21	20.83	2.18 × 10^7^		0.08			Negative	
37	NA°	NA°	1					
67	18.99	7.80 × 10^7^	2	0.14	0	0.05	Negative	
82	18.14	1.41 × 10^8^						
111	21.75	1.15 × 10^7^	3	-	-	-	-	-
124	19.71	4.74 × 10^7^	4					
137	28.62	9.85 × 10^4^	5	0.19	0	0.12	Borderline Positive	>250
152	35.20	1.03 × 10^3^		0.21			Negative	
161	35.72	7.18 × 10^2^		0.41				

* RT-qPCR cycle threshold (Ct)-value of SARS-CoV-2 N1-gene; ** Semi-quantitative estimation of SARS-CoV-2 viral load calculated from the standard curve (cfr. Method Section); *** Whole genome sequencing time points; **** SARS-CoV-2 anti-nucleocapsid protein antibodies; ***** SARS-CoV-2 anti-spike protein antibodies; ° Due to low specimen volume, no RT-qPCR was performed.

**Table 2 viruses-14-00752-t002:** Nucleotide and amino acid comparison of the patient’s SARS-CoV-2 genomes to the reference genome Wuhan-Hu-1.

			ORF1a										ORF1b		S									ORF3a	ORF8	ORF9b	N			
Sequence Name	Clade	#Nt	241 *	3037 *	5178	5184 ^¥^	5807	5869	6031	8748	10279	12747	14408	20742	21846	21990–21995	22287–22295	22985	23016	23031	23063	23114	23403 *	26109	27992	28321	28830	28881	28882	28883
		#AA																												
IC|S01|d37	20B GR	14	T	T	C	T	A	C	T	C	C	A	T	C	C	TTTATT	CTTGCTTTA	G	G	T	A	C	G	T	C	T	C	A	A	C
		10																												
IC|S02|d67		15	T	T	T	C	A	C	T	C	C	A	T	C	T	TTTATT	---------	G	G	T	A	C	G	T	C	T	C	A	A	C
		11																												
IC|S127|d111		20	T	T	T	C	G	T	T	C	C	A	T	C	T	TTTATT	---------	G	T	T	T	C	G	T	C	T	T	A	A	C
		15																												
IC|S128|d124		21	T	T	T	C	G	T	T	C	C	A	T	C	T	------	---------	T	G	C	T	C	G	T	C	T	T	A	A	C
		16																												
IC|S129|d137		24	T	T	T	C	G	T	T	T	T	A	T	C	T	------	---------	T	G	C	T	G	G	T	C	T	T	A	A	C
		20																												
Amino acid change					T1638I	P1640L	I1848V			A2828V		T4161N	P314L	Q2425H	T95I	143-4ΔY145D	241-3Δ	A475S	G485V	F490S	N501Y	L518V	D614G*	E239D	I10T	R13L	S186F	R203K		G204R

* Mutations defining GISAID’s G clade; ¥ Within the quasispecies of all samples, both C and T occur at this position. #Nt = number of nucleotide changes compared to Wuhan-Hu-1; #AA = number of amino acid changes compared to Wuhan-Hu-1. Clade = respectively Nextstrain (20B); GISAID (GR) nomenclature. IC = immunocompromised patient; d = day; ORF = open reading frame; M = membrane; S = spike; N = nucleoprotein.

## Data Availability

Not applicable.

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
