# Peer review of "Viral Evolution and Immunology of SARS-CoV-2 in a Persistent Infection after Treatment with Rituximab"

_viruses, 2022, doi:10.3390/v14040752_

Round 1
Reviewer 1 Report
The manuscript entitled “Viral Evolution and Immunology of SARS-CoV-2 in a Persis-2 tent Infection after Treatment with Rituximab” has introduced a case report with a lymphoma that had received therapy with rituximab and remained SARS-CoV-2 RT-qPCR positive for 161 days. The report has summarized the clinical course especially the SARS-CoV-2 RT-qPCR and genome mutation results during 161 days. The results have provided some useful reference for the treatment of SARS-CoV-2 in immunocompromised individuals.
However, there are some shortages in the manuscript:
- figure1: the usage detail of convalescent plasma, Dexamethasone and antibiotics may be added to figure 1 and result part, including dosage, frequency and course.
- figure2 is not necessary in the manuscript.
- table2: there are some mistakes in the table- amino acid change.
- figure4: please add the numerical value of different lymphocytes to the figure.
Author Response
Figure1: the usage detail of convalescent plasma, Dexamethasone and antibiotics may be added to figure 1 and result part, including dosage, frequency and course.
Thank you for this valuable remark. Names of the antibiotic regimens were added to Figure 1, dose and course duration were added to the text. We tried adding dosage and duration to Figure 1, but this rendered the figure less clear.
Figure2 is not necessary in the manuscript.
Thank you for this remark, figure 2 was deleted from the manuscript.
Table2: there are some mistakes in the table- amino acid change.
Thank you for this important correction, the table was adapted.
Figure4: please add the numerical value of different lymphocytes to the figure.Thank you for this remark, the numerical values were added to Figure 4.
Reviewer 2 Report
The manuscript entitled, "Viral evolution and immunology of SARS-CoV-2 in a persistent infection after treatment with rituximab" is a well-described case study in which an immunocompromised patient with underlying conditions maintained an active SARS-CoV-2 infection for 161 days. The manuscript describes the accumulation of mutations via prolonged shedding of SARS-CoV-2, which has both clinical and viral evolutionary significance.
Major comments:
Suggest adding a sentence describing the drug rituximab in the introduction, including it's mechanism of action and clinical use.
Figure 5 is not described anywhere in the text and the legend is poorly descriptive. Figure should either be addressed in the text or removed altogether.
Minor comments
Line 53: suggest removing the word "breeding" from this sentence.
Line 90: Replace "there was" with "it was"
Figure 3: What do the pink highlighted regions with dashed lines indicate?
There are two different font sizes in the discussion, please resolve.
Author Response
Major comments
Suggest adding a sentence describing the drug rituximab in the introduction, including it's mechanism of action and clinical use.
Thank you for this remark, a sentence on rituximab action and use was added to the introduction. (line 56-59)
Figure 5 is not described anywhere in the text and the legend is poorly descriptive. Figure should either be addressed in the text or removed altogether.
Thank you for this remark. A reference to figure 5 was added to the manuscript (Line 176) and the legend was elaborated.
Minor comments
Line 53: suggest removing the word "breeding" from this sentence.
Thank you for this suggestion, the word ‘breeding’ was deleted. (line 53)
Line 90: Replace "there was" with "it was"
The adjustment was made. (line 93)
Figure 3: What do the pink highlighted regions with dashed lines indicate?
Thank you for this valuable question, the highlighted regions are depletions, this information was added to the figure legend.
There are two different font sizes in the discussion, please resolve.
Thank you for this correction, the font size was adapted.
Reviewer 3 Report
Van der Moeren and colleagues present an interesting case study of an immunocompromised individual with prolonged SARS-CoV-2 infection which resulted in detectable within-host evolution. It is a well written case report and adds to the growing literature on the significance ongoing viral evolution and persistence. There were only a few comments that I wanted to bring forth:
- Is my assumption that the lack of B/T cells is due to the Rituximab treatment? And if so, how long after would you expect B cells to return? Besides Rituximab, was there any other regiment the patient was on?
- For language, it might be better to state that there were no detectable B-cells instead of zero since all assays have some limit of detection. Of interest, would you have been able to measure the abundance of plasma cells?
- In the abstract and later in the report, the authors state phylogenetic analysis was done, however, a tree is never shown. Like in many other case reports, it is of this Reviewers' opinion that generating a model-fit maximum-likelihood tree with the consensus sequences at each time point for this patient along with other case report sequences/references would be beneficial to this study. This would allow the reader to observe the direct within-host evolution that is further described.
- Are the statistics done to demonstrate that T cells and mDCs were significantly reduced in the patient compared to control? And maybe as well as B cells, but it is hard to tell by eye.
- In figure 5, it is hard to know if nay of the results are significant. Based on literature the data is consistent, but is otherwise hard to interpret. Maybe more time is required in the results?
Author Response
Is my assumption that the lack of B/T cells is due to the Rituximab treatment? And if so, how long after would you expect B cells to return? Besides Rituximab, was there any other regiment the patient was on?
Thank you for these important questions.
As rituximab is an anti-CD20 antibody targeted against B-lymphocytes the B-cell depletion is expected to be maily due to the rituximab use. Rituximab is also known to induce substantial T-cell depletion (Mélet J, Mulleman D, Goupille P, Ribourtout B, Watier H, Thibault G. Rituximab-induced T cell depletion in patients with rheumatoid arthritis: association with clinical response. Arthritis Rheum. 2013 ; Ramwadhdoebe T, Boumans M, Bruijnen S, et alA8.10 The effect of rituximab treatment on B and T cell subsets in lymphoid tissues of patients with rheumatoid arthritisAnnals of the Rheumatic Diseases 2015).
Time to immune reconstitution varies after rituximab administration, but is usually 3-6 months in this patient population (although it can persist upto 12 months).
At the time of presentation rituximab was the only B-cell depleting treatment, the patient had received bendamustin (anti B and T-lymphocyte) treatment 3 months before presentation. The effect of bendamustin on the lymphocytes at this points is likely to be worn of but we cannot exclude a small remaining effect. A short explanation on rituximab was added to the manuscript introduction. (line 56-59)
For language, it might be better to state that there were no detectable B-cells instead of zero since all assays have some limit of detection. Of interest, would you have been able to measure the abundance of plasma cells?
“zero B-cells” was replaced by “no detectable B-cells”. (line 99-100) The abundance of plasma cells was detectable in some but not all samples.
In the abstract and later in the report, the authors state phylogenetic analysis was done, however, a tree is never shown. Like in many other case reports, it is of this Reviewers' opinion that generating a model-fit maximum-likelihood tree with the consensus sequences at each time point for this patient along with other case report sequences/references would be beneficial to this study. This would allow the reader to observe the direct within-host evolution that is further described.
Thank you for this very valuable remark, we are sorry to report there was no phylogenetic analysis performed. This was a misinterpretation of the first author. The statement on the performance of phylogenetic analysis was deleted from the manuscript. We sincerely apologize for this error.
The autors discussed adding a phylogenetic tree. We believe that table 2 gives a sufficiently clear oversight of the observed mutations and as we have the dates the sequences were obtained on, it is clear which sequences share an ancestor.
Furthermore, we were concerned the algorithms we use for phylogenetic analysis might lead to false conclusions when used for the evaluation of in-host evolution. Nevertheless, we did start to run a phylogenetic analysis - it was however not ready at the moment of the deadline for resubmission of the manuscript.
We greatly appreciate the valuable suggestion but hope the reviewer can accept the above argumentation.
Are the statistics done to demonstrate that T cells and mDCs were significantly reduced in the patient compared to control? And maybe as well as B cells, but it is hard to tell by eye.
Thank you for remark, we added significance levels analysed by Mann-Whitney test to Figure 4.
In figure 5, it is hard to know if nay of the results are significant. Based on literature the data is consistent, but is otherwise hard to interpret. Maybe more time is required in the results?
Thank you for this very valuable remark, we added significance levels analysed by Mann-Whitney test to Figure 5 and adapted the text accordingly.
Round 2
Reviewer 3 Report
I would like to thank the authors of this Case Report for addressing my comments. It is a very nice case and exemplifies the intra-host evolution in immunocompromised individuals. I agree that the ML tree does not necessarily add to the message. Leaving in as supplementary I think would suffice if you decide to include it. It does reinforce your Table but also illustrates the time progressive evolution which some in the field would appreciate. Brilliant work.